# The Cognitive Sequelae of Transient Ischemic Attacks—Recent Insights and Future Directions

**DOI:** 10.3390/jcm11092637

**Published:** 2022-05-07

**Authors:** Aravind Ganesh, Philip A. Barber

**Affiliations:** 1Calgary Stroke Program, Departments of Clinical Neurosciences, University of Calgary Cumming School of Medicine, Calgary, AB T2N 4N1, Canada; aganesh@ucalgary.ca; 2Department of Community Health Sciences, University of Calgary Cumming School of Medicine, Calgary, AB T2N 4N1, Canada; 3Hotchkiss Brain Institute, University of Calgary Cumming School of Medicine, Calgary, AB T2N 4N1, Canada; 4Department of Radiology, University of Calgary Cumming School of Medicine, Calgary, AB T2N 4N1, Canada

**Keywords:** transient ischemic attack, cognition, dementia, cognitive impairment

## Abstract

There is now considerable evidence that Transient Ischemic Attack (TIA) carries important sequelae beyond the risk of recurrent stroke, particularly with respect to peri-event and post-event cognitive dysfunction and subsequent cognitive decline. The occurrence of a TIA could provide an important window in understanding the relationship of early mixed vascular-neurodegenerative cognitive decline, and by virtue of their clinical relevance as a “warning” event, TIAs could also furnish the opportunity to act preventatively not only for stroke prevention but also for dementia prevention. In this review, we discuss the current state of the literature regarding the cognitive sequelae associated with TIA, reviewing important challenges in the field. In particular, we discuss definitional and methodological challenges in the study of TIA-related cognitive impairment, confounding factors in the cognitive evaluation of these patients, and provide an overview of the evidence on both transient and long-term cognitive impairment after TIA. We compile recent insights from clinical studies regarding the predictors and mediators of cognitive decline in these patients and highlight important future directions for work in this area.

## 1. Introduction

Transient ischemic attack (TIA), by definition, is caused by transient ischemia of the brain resulting in acute focal neurological symptoms that resolve within 24 h [1]. There is now considerable evidence that TIA carries important sequelae beyond the risk of recurrent stroke, particularly with respect to peri-event and post-event cognitive dysfunction and subsequent cognitive decline [2]. The concept of vascular cognitive impairment (VCI) is defined by evidence of cerebrovascular disease either in isolation or in combination with neurodegenerative disease [3], and vascular dementia has generally focused on the development of dementia following clinically-evident strokes or the radiological accumulation of brain infarcts [4]—but how do TIAs fit into this picture? This question is important for both clinicians and researchers in the overlapping fields of stroke and dementia, given the persistently high incidence of TIAs at a population level (about 1.19/1000 person-years in the Framingham Heart Study, 1948–2017) [5].

It is clear cerebrovascular disease—be it in the form of strokes or cerebral small vessel disease—is a crucial mediator of cognitive decline [6], with vascular contributors accounting for at least a third of all dementia [7,8]. There is also growing evidence that even in more typical “neurodegenerative” cognitive decline, such as that attributed to Alzheimer’s Disease (AD), vascular factors play a key pathological role [9]. TIAs could provide an important window into what is going on in early VCI or cognitive decline and, by virtue of their clinical relevance as a “warning” event, could also furnish the opportunity to act preventatively not only for stroke prevention but also for dementia prevention. Recently, parts of the world that have demonstrated a declining incidence of stroke have also reported a declining incidence of dementia, potentially reflecting the impact of such preventative care [10].

In this review, we discuss the current state of the literature regarding the cognitive sequelae associated with TIA, review important challenges in the field, compile recent insights from clinical studies, and highlight the important future directions for work in this area.

## 2. Definitional and Methodological Challenges

The study of cognitive sequelae of TIA is complicated by two major definitional challenges, namely, challenges with defining (a) what constitutes a TIA and (b) what constitutes cognitive sequelae. The definition of TIA has changed several times in the past four decades [11,12,13]. In clinical practice and in population-based studies, both time-based and tissue-based definitions of TIA are in use [14,15]. In part, this reflects the practical realities of varying access to MRI as part of the TIA evaluation process, as well as the fact that many patients still do not come to medical attention immediately after a TIA-like event; the likelihood of finding a diffusion-weighted imaging (DWI) lesion on MRI diminishes with the number of days after a minor stroke [16]. Additionally, as imaging techniques become increasingly advanced, the detection of a cerebral infarct associated with a TIA-like event will end up hinging on the sensitivity of the neuroimaging equipment used. For example, a 3T MRI can be expected to pick up more infarcts than a 1.5T MRI, which in turn will be more sensitive than a 0.15T MRI, which is more sensitive than a CT scan [17]. Now, even 7T and 11T superconducting magnets are available. Thus, the application of a strict tissue-based definition of TIA would mean that the patient’s diagnosis can change depending on the scanner they have access to; this is an important limitation to the generalizability of that definition. To overcome this definitional challenge, many studies now use the term “TIA/minor stroke”, which seems justifiable considering that TIA and minor stroke are ultimately on a comparable “mild” band of the stroke severity spectrum [17,18,19,20,21,22,23]. The upper threshold of stroke severity for the “minor stroke” definition has been variably defined, generally being in the range of 3–5 points on the National Institutes of Health Stroke Scale (NIHSS) [19,24,25].

As for the definition of what constitutes the cognitive sequelae of a TIA, this requires us to first establish what the cognitive baseline was for the patient. Older patients, patients with prior functional impairment, and multiple comorbidities are more likely to have pre-existing cognitive decline or dementia, which (if not captured) can confound the determination of post-event cognitive decline [26]. Obtaining a collateral history from an informant—someone who lives with the patient or knows them well—is generally key to identifying pre-existing issues. The 16-item IQCODE (Informant Questionnaire on Cognitive Decline in the Elderly) is one example of an informant-completed tool that demonstrates excellent accuracy in detecting pre-existing dementia in TIA/stroke patients. In the population-based Oxford Vascular Study (OXVASC), the optimal cut-off score for the IQCODE to identify pre-existing dementia was >3.48 (sensitivity 89.7%, specificity 84.2%) [27]. The greatest differences between dementia and no-dementia groups were seen in questions relating to finances, using gadgets, arithmetic, and learning new things.

## 3. Evaluation of Cognitive Impairment after TIA

Next is the problem of ascertaining cognitive impairment after a TIA. As patients themselves may not link cognitive issues to their TIA, and the changes might be subtle, this requires us to formally assess cognition. Several “global” cognitive test assessments are commonly performed in practice or in research settings to evaluate cognition after TIA or stroke. Commonly used tests include the Mini-Mental State Examination (MMSE) [28], Montreal Cognitive Assessment (MoCA) [29], and the longer Addenbrooke’s Cognitive Examination-Revised (ACE-R) [30].

These cognitive tests have differing sensitivities for cognitive impairment. For example, the MoCA is more sensitive to mild cognitive impairment (MCI) than the MMSE, with over half of TIA patients with a “normal” score (≥27) on the MMSE 6 months or more post-event, scoring < 26 on the MoCA [31,32,33]. In OXVASC, the MoCA picked up substantially more cognitive abnormalities after TIA than the MMSE at both 6-month and 5-year follow-up [34]. This might be due to the detection of different patterns of cognitive domain impairment. An interesting study in this regard compared performance on the MMSE and MoCA among consecutive patients without major cognitive impairment (i.e., excluding those with an MMSE score of <24) in two different cohorts, a TIA/stroke cohort from OXVASC and a memory research cohort, the Oxford Project to Investigate Memory and Ageing [31]. The stroke patients had the lowest cognitive scores, whereas TIA and memory cohort patient scores were similar. The rates of MoCA score in <26 in subjects with “normal” MMSE (≥27) were lowest in memory subjects, intermediate in TIA, and highest after stroke (34% vs. 48% vs. 67%). The TIA/stroke patients scored lower than the memory cohort patients on all MoCA frontal/executive subtests, with differences being most marked in visuoexecutive function, verbal fluency, and sustained attention. Stroke patients performed worse than TIA patients only on MMSE orientation in contrast to six out of ten subtests of the MoCA. Thus, the MoCA captured more differences in cognitive profile between TIA, stroke, and memory research subjects than the MMSE, including among patients with a “normal” MMSE.

VCI, broadly speaking, is known to have a preponderance of executive dysfunction, including slowed information processing, impaired set-/task-shifting, and deficits in working memory [35,36]. Commonly-used cognitive tests again differ considerably in their sensitivity for such dysfunction in the setting of TIA or minor stroke. For example, in OXVASC, all three visuo-executive sub-tests of the MoCA (trails, cube copying, and clock drawing) detected more patients with visuo-executive function than the pentagon copying sub-test of the MMSE, resulting in an over 10-fold superiority of the MoCA in identifying visuo-executive dysfunction (odds ratio 11.4, 95%CI 8.2–15.8) [37]. However, even measures such as the MoCA have more limited sensitivity, with specificity in the range of 70%, in identifying subtle cognitive changes particularly among younger or otherwise healthy patients, and might suffer from a ceiling effect [38]. To help assess domains commonly affected by VCI with greater granularity, the National Institute of Neurological Disorders and Stroke (NINDS) and the Canadian Stroke Network (CSN) have proposed 5-min, 30-min, and 60-min cognitive testing protocols as part of the NINDS-CSN VCI Harmonization Standards [39]. In a sample of 100 consecutive OXVASC patients tested one year or more after TIA/stroke (mean age 73.4 years, 44% female, 44% TIA), 42% had MCI, which was defined as impairment (1.5 standard deviations or more) in one or more domains on the NINDS-CSN VCI battery compared with age- and education-matched published norms, without impairment of functional activities of daily living [40]. Most patients with MCI had non-amnestic single domain MCI (*n* = 19), followed by amnestic multiple domain (*n* = 10), and non-amnestic multiple domain (*n* = 9). Sensitivity and specificity for MCI were optimal with MoCA < 25 (sensitivity = 77%, specificity = 83%) and ACE-R < 94 (sensitivity = 83%, specificity = 73%), but both tests detected amnestic MCI better than non-amnestic single-domain impairment. The MMSE demonstrated a ceiling effect and only achieved sensitivity > 70% at a cut-off of <29. The lack of timed measures of processing speed may explain the relative insensitivity of the MoCA and ACE-R to single non-amnestic impairment.

In-person follow-up is often not possible for many patients after TIA. In such patients, telephone-based cognitive assessments are a potentially helpful alternative. Two tests that have been specifically studied in the setting of TIA/stroke are the telephone Montreal Cognitive Assessment (T-MoCA: MoCA items not requiring pencil and paper or visual stimulus) and the modified Telephone Interview of Cognitive Status (TICSm) [41]. In OXVASC, optimal cut-offs for diagnosing MCI were 18 to 19 (out of 22) for T-MoCA and 24 to 25 (out of 39) for TICSm [42]. In the German multi-center Determinants of Dementia After Stroke study, 96 stroke survivors completed both face-to-face comprehensive neuropsychological testing (at 6, 12, 36, and 60 months) and the Clinical Dementia Rating Scale, which were used as reference standards, as well as the TICS and T-MoCA at two separate telephone visits, 1 week before and 1 week after each face-to-face visit [43]. The optimal cut-off for multi-domain MCI in this study relative to the Clinical Dementia Rating Scale was 19 for T-MoCA (similar to OXVASC) but much higher for TICSm at 36. Overall, both T-MoCA and TICSm were found to be feasible and valid telephone tests of cognition after TIA/stroke but generally performed better in detecting multi-domain vs single-domain impairment. Importantly, T-MoCA is limited in its ability to assess visuoexecutive and complex language tasks compared with face-to-face MoCA.

Furthermore, in the 15 years since the paper-based batteries of the NINDS-CSN VCI Harmonization Standards were proposed [39], computerized and web-based testing have gained considerable support among researchers, with numerous studies demonstrating the feasibility and reliability [44,45,46,47,48] of completing cognitive tasks on computers, tablets, or phones in patients with concussion [49,50], MCI [51,52], multiple sclerosis [53,54] and dementia [44,55]. The Cambridge Brain Sciences (CBS) battery is one example in this field, currently being used in the Alteplase compared to the Tenecteplase trial in patients with ischemic stroke (NCT03889249). Unlike traditional paper-based tests, the multi-domain tests on computerized batteries such as the CBS increase or decrease in difficulty based on the participants’ performance. This dynamicity helps the tests achieve greater sensitivity than traditional paper tests [56,57]. These tests also have the advantage of having normative data from very large numbers of people; for example, more than eight million CBS tests have been taken to date, and comprehensive normative data from 75,000 healthy participants are available [58]. Most of these individuals played every test at least twice, and more than 5000 have played every test more than 10 times, creating reliable measures of variability used to discriminate meaningful changes in performance. As the population experiencing TIAs is expected to become increasingly computer-savvy in the coming years, these tests present a unique opportunity for more granular, personalized testing of cognition after TIA. However, these tests are also not without their limitations; for example, the CBS emphasizes attentional capacity but does not assess verbal or visual memory well.

Once we have decided what test to use to evaluate cognition after a TIA, the next challenge becomes defining what score constitutes cognitive impairment. Each test has its own proposed cut-off for the populations with cognitive complaints in whom they were first tested, but these may not apply in the setting of TIA. In OXVASC, different cut-offs (>1, >1.5, and >2 standard deviations) on a given test relative to published norms were compared together with the use of single versus multiple tests (MMSE and/or MoCA) to define domain impairment [59]. The rates of cognitive impairment ranged from 15% when defined using a cut-off of >2 standard deviations to 67% when using a cut-off of >1 standard deviation, over a four-fold variation in cognitive impairment estimates. Thus, studies of cognitive impairment after TIA can end up with vastly different risk estimates depending on the threshold set for defining impairment. This also has implications for sample size calculations for future randomized trials seeking to address such impairment. Consensus definitions are needed to facilitate greater uniformity in study design.

## 4. Confounding Factors in the Evaluation of Cognition after TIA

Perhaps the biggest challenge in evaluating the cognitive sequelae of TIAs is that of disentangling changes attributable to the TIA itself versus other confounding factors. Some of these factors directly complicate the cognitive assessment of patients. For example, fatigue is common after TIA. More than one in four patients have fatigue after TIA (versus more than one in two after stroke) [22]. Fatigue can interfere with cognitive testing, leading patients to perform worse than controls, who are relatively free of fatigue [22]. In practice, this means that factors such as the length of the visit and the time of day can affect the cognitive performance of patients after TIA, particularly in older patients. In the OXVASC study, the proportion of patients ≥ 70 years who appeared to have a MoCA score < 20 in clinic-based testing was significantly greater in the afternoon than in the morning (10.9% versus 1.8%) with no differences based on age, education, disability, or vascular risk factors [60]. These findings imply that the timing of cognitive assessments should be considered in the evaluation of patients after TIA.

Depression is another challenge. Depression occurs commonly after TIA and minor stroke and, similar to fatigue, may influence performance on cognitive testing [61]. For example, in a cohort study of 82 patients with TIA/minor stroke, 43.9% of patients were diagnosed with depression at 10 days post-event according to the criteria described in the Diagnostic and Statistical Manual of Mental Disorders, Fifth Edition, while 34.1% had apathy [62]. At 12 months, 35.7% of patients still had apathy, but only 8.6% still had depression. Several scores were associated with a diagnosis of depression and apathy, including the Beck Depression Inventory-II [63] score and the Montgomery Asberg Depression Rating Scale [64]. Such scores may help adjust for these symptoms in studies of post-TIA cognitive decline.

Another important issue is attrition—that is, the loss of patients to follow-up after an event such as a TIA, which can particularly skew estimates of the prevalence of cognitive sequelae. For example, in OXVASC, 42% of patients who developed dementia after TIA or stroke had been unavailable for face-to-face clinic follow-up at the time of their dementia diagnosis, which was instead captured in that study using a combination of home visits, telephone follow-up, and primary care records [65]. The five-year cumulative incidence of post-event dementia was only 17% in clinic-assessed patients versus 45% in non-clinic-assessed patients. This study demonstrated that the exclusion of patients unavailable for clinical follow-up results in an underestimation of the risk of post-event dementia. Studies of cognitive sequelae after TIA should ideally use multiple follow-up methods to improve ascertainment of long-term cognitive outcomes.

There is also the issue of patients who are “untestable”—substantial numbers of patients are simply not going to be testable with a given cognitive test. For example, in OXVASC, 12% of 378 patients with TIA were untestable at baseline [66]. While this was far better than rates of untestable patients with stroke at baseline (54%), given the absence (by definition) of persistent neurological deficits, such as aphasia with TIA, it is important to note that being untestable was associated with older age and pre-morbid dependency, as well as with pre-existing dementia. There is a paucity of high-quality data on similar issues in more diverse cohorts (OXVASC consisting almost entirely of White English-speaking patients), as in practice, issues of language and literacy are likely to impede cognitive testing. Many cognitive tests are also subject to cultural biases. Strategies to mitigate these issues could include offering testing in different languages, which is easier to do with computerized testing. Studies may also consider using less culturally biased tests, such as the Rowland Universal Dementia Assessment Scale, which was specifically created to test for dementia in culturally and linguistically diverse populations [67]. However, it is yet to be formally validated in the setting of VCI.

The key confounding factors and possible mitigating strategies are summarized in Table 1.

## 5. Transient Cognitive Impairment

Transient cognitive impairment (TCI) was first described in patients with acute major stroke and is described as acute cognitive deficits occurring early after a stroke that recover to at least some extent, although cognitive recovery does not necessarily parallel physical post-stroke recovery [71,72]. TCI may be seen as being on a similar spectrum as delirium, which is common after hospitalized stroke and strongly associated with the subsequent development of dementia [73,74,75]. TCI was studied in OXVASC in the setting of TIA/minor stroke (2002–2005). MMSE was performed in consecutive testable patients with TIA/minor stroke seen acutely (1–7 days) and after 7 days, with TCI being defined as a baseline MMSE score ≥ 2 points below the 1-month follow-up score [76]. Cognitive impairment was defined as a MoCA score < 26 at 1-year to 5-year follow-up. Among 280 patients, TCI was more frequent in those seen at 1 to 7 days post-event (38.9%) versus later (19%) or in non-cerebrovascular patients (21%). Among TIA patients, TCI was associated with acute confusion and persisted beyond the resolution of focal symptoms, while in minor stroke, TCI was also associated with acute infarct on CT and residual focal deficits. Whereas patients with TCI had similar MMSE scores by 1-month compared to those without TCI, they had a higher 5-year risk of cognitive impairment and dementia.

TCI was also evaluated in a prospective cohort study of 100 patients with TIA/minor stroke in Edmonton, Canada, without a history of cognitive impairment, assessed within 72 h of onset, who underwent repeated testing with the MMSE and MoCA at days 1, 7, 30, and 90 post-event [77]. Cognitive impairment was defined as an MoCA of < 26 and an MMSE of ≤ 26, cognitive impairment was detected in 54% of patients with MoCA and 16% with the MMSE at baseline; the MoCA scores improved on days 7, 30, and 90, and this related to the resolution of recall deficits.

In a subsequent study, 326 patients from OXVASC with TIA/minor stroke were evaluated with transcranial Doppler and blood pressure monitoring [78]. Patients were classified as having TCI (baseline MoCA < 26 with ≥2 points increase at 1-month), persistent MCI (MoCA < 26 with <2 points increase), and no cognitive impairment (MoCA ≥ 26). Patients with TCI had higher systolic blood pressure and lowered cerebral blood flow velocities, particularly end-diastolic velocity and mean flow velocity on transcranial doppler than those with no cognitive impairment, but had similar clinical and hemodynamic profiles to those with persistent MCI. In all three groups, systolic blood pressure fell between baseline and 1 month (mean reduction 14.01 ± 21.26 mmHg) while end-diastolic velocity and mean flow velocity increased (mean increase 2.42 ± 6.41 and 1.89 ± 8.77 cm/s, respectively). This study demonstrates that patients with TCI have a similar clinical and hemodynamic profile as patients with more persistent cognitive changes, but this phenomenon does not appear to be due to exaggerated acute reversible global hemodynamic changes.

## 6. Long-Term Cognitive Decline after TIA

Studies examining the prevalence of long-term cognitive decline after TIA have suffered from different types of confounding, detection bias, and attrition [79]. The prevalence of post-TIA MCI is in the range of 29–68% [79]. Patients evaluated with shorter cognitive screening tools (such as MMSE or MoCA) have generally been substantially older than those who have undergone a full neuropsychological assessment. The cognitive profile has generally consisted of prominent executive deficits.

In OXVASC, among 688 patients with TIA, 5.2% had dementia at 1-year post-event (compared to 8.2% of those with minor stroke and 34.4% of those with severe stroke) [6]. Compared with the United Kingdom-based age-matched and sex-matched population, the 1-year standardized morbidity ratio for the incidence of dementia was 3.5 (95%CI 2.5–4.8), with the prevalence of dementia in 1-year TIA survivors being brought forward by 2 years.

In a case-control study of 107 patients with TIA and no prior stroke (63% women, mean age 56.6 years, tested within 3 months of the event) versus 81 controls (56% women, mean age 52.9) [2], the patients with TIA performed worse on all cognitive domains in a comprehensive neuropsychological assessment, except episodic memory. In particular, they had a 22.5-fold higher odds of impairment in working memory, 6.8-fold higher odds of impairment in attention, and 7.1-fold higher odds of impairment in processing speed. Over 35% of patients with TIA had impairment in one or more cognitive domains within 3 months of their event, with impairment defined as a domain z-score < −1.65 (derived based on the mean of the control group). In another study of 39 consecutive patients with carotid occlusion and ipsilateral TIAs (tested 2–6 months after the event, median 54 days) and 46 healthy controls, all of whom underwent neuropsychological testing, 54% of the patients were cognitively impaired [80].

In a prospective cohort study of 121 patients with TIA and Transient Neurological Attacks [81] in the Netherlands, the patients underwent comprehensive cognitive assessment and MRI within 7 days of the event, with cognitive testing repeated after 6 months [82]. Executive function performance decreased over time, whereas attention improved, and information processing speed and episodic memory remained unchanged. Some of these TIA patients would have been classified as an ischemic stroke as 26% had a DWI lesion; in this regard, patients with a DWI lesion had persistently worse executive function than those without a DWI lesion. In a study of 103 patients with TIA and Transient Neurological Attacks by the same group, subjective cognitive complaints and fatigue both increased in severity from baseline to 6 months only among patients with DWI lesions [83].

Few studies have addressed cognitively relevant motor performance. In a recent cohort study of 48 individuals with TIA assessed at 2, 6, 12, and 52 weeks after symptom resolution on behavioral tasks with the Kinarm Exoskeleton robot, up to 51.3% demonstrated an impairment on a given task within 2-weeks of symptom resolution, and up to 27.3% had a persistent impairment after 1 year, with about a quarter of patients having statistically significant deteriorations in performance after 1 year [84]. The impairments were mostly in tests of cognitive-motor integration.

In summary, several cognitive domains have been reported to be affected in patients after TIA, mostly related to different aspects of executive function and related frontal-subcortical circuit functions (Table 2). However, it is important to emphasize that prior cohort and case-control studies have tended to rely on cognitive screening tests, such as MoCA—limiting their ability to deeply characterize impairment patterns—or may have used neuropsychological batteries more heavily weighted towards assessing domains such as executive function, in turn feeding into the perception that VCI is characterized predominantly by weakness in these domains. When visual and verbal memory have been assessed in detail, studies have found impairments in these domains as well. For example, in a study of 95 patients with TIA (with a 90-day follow-up assessment) and 51 non-TIA controls in the Predementia Neuroimaging of Transient Ischemic Attack (PREVENT) study, TIA subjects had lower scores on the Brief Visuospatial Memory Test-Revised and on verbal memory tests including the Auditory Verbal Learning Test by the World Health Organization and the University of California, Los Angeles, and the ACE-R verbal memory tasks [85]. Overall, the extent and quality of available evidence for characterizing the cognitive profile of patients after TIA remains limited.

## 7. Predictors and Mediators of Cognitive Decline after TIA

Studies of cognitive decline after TIA have suffered from considerable heterogeneity and insufficient data limiting conclusions about potential causative factors [79]. The complex interplay of potential contributory factors or mediators of cognitive decline (as well as some potential confounders) is shown in Figure 1.

Age is by far the most consistent risk factor for cognitive decline after TIA and the least surprising. The 5-year risk of post-TIA dementia in OXVASC was associated with age, event severity, previous stroke, dysphasia, baseline cognition, low education, pre-morbid dependency, leukoaraiosis, and diabetes [6]. Early cognitive impairment is also an important predictor of future dementia—in OXVASC, a score of less than 24 on the MMSE at baseline was among the strongest predictors of 5-year dementia risk, emphasizing the prognostic usefulness of early cognitive testing after TIA/stroke.

The burden of white matter hyperintensities (WMH) or leukoaraiosis is recognized as an important imaging marker of brain frailty and is perhaps the most extensively studied imaging marker. Impairment on cognitive tests such as the MoCA, several months after a TIA, is strongly associated with the presence of leukoaraiosis [86]. In OXVASC, worse MoCA and MMSE scores 1-month after TIA/stroke were significantly correlated with higher WMH volumes and reduced fractional anisotropy in almost all white matter tracts, with early cognitive impairment detected with the MoCA being independently correlated with these measures [87]. In a prospective cohort study of 115 patients with TIA/minor stroke (NIHSS ≤ 3) recruited within 72 h of onset in Alberta, Canada, with cognitive testing at baseline, 7-days, 30-days, and 90-days post-event, cognitive impairment rates (defined as MoCA score < 26) were similar in patients with (47/91, 52%) and without diffusion-weighted imaging lesions (13/24, 54). However, persisting impairment at 30-days was correlated with WMH volumes, suggesting that subclinical cognitive impairment and/or impaired ability to compensate for the effects of acute ischemic infarcts may be at play [25].

WMHs generally increase with age. In OXVASC, among 566 patients with TIA or minor stroke, WMH volumes and MRI measures of white matter disruption (mean diffusivity and fractional anisotropy) were strongly associated with cognitive status in patients aged ≤ 80 years (all *p* < 0.001) but not in those > 80 years (not significant for any of these measures) [23]. In particular, lower MoCA scores were associated with frontal WMH in patients ≤ 80 years but not >80 years.

This suggests that MRI markers of white matter damage are more important in younger ages than in older ages with respect to cognitive status. However, WMHs on conventional MRI only correlate weakly with cognition and dementia risk; Diffusion Tensor Imaging (DTI) may be a more sensitive marker of white matter disease. DTI can detect microstructural disease in TIA patients before cognitive symptoms develop. For example, in a recent analysis of 95 consecutive TIA patients and 51 controls from PREVENT, the TIA group exhibited higher mean diffusivity values in the fornix and lower fractional anisotropy in the superior longitudinal fasciculus, genu, and uncinate fasciculus [85]. TIA patients scored lower on the ACE-R, with lower scores in memory and processing speed but showed no differences in overall MoCA and MMSE scores in this study. Lower fractional anisotropy and higher mean diffusivity in the fornix, superior longitudinal fasciculus, and uncinate fasciculus were associated with poorer performance on tests of visual memory and executive function but not verbal memory. Lower fractional anisotropy in the uncinate fasciculus and fornix were related to higher timed scores on the Trail Making Test-B (TMT-B), and higher superior longitudinal fasciculus mean diffusivity was related to higher scores on TMT-B, confirming worse executive performance in the TIA patients.

WMHs are only one imaging feature of small vessel disease (SVD), other features include lacunes, microbleeds, and enlarged perivascular spaces [88]. In a study of 234 TIA/stroke patients captured in the Virtual International Stroke Trials Archive, evaluated with the MMSE at 1 year follow-up, the features of SVD were independently associated with mesial temporal atrophy (MTA), which was seen to a moderate or severe degree in 44% of patients [89]. After adjusting for age, sex, disability, hypertension, and diabetes mellitus, MTA was the only radiological feature independently associated with cognitive impairment (both when defined as MMSE ≤ 26 or ≤23). This is an interesting finding since MTA is more a feature of AD than VCI, pointing to the challenge of disentangling the contributions of comorbid neurodegenerative disease from cerebrovascular contributors to post-TIA cognitive decline.

Brain atrophy after TIA is an area of ongoing study. In one study of 60 patients with a first, isolated TIA and 26 age- and sex-matched controls, all of whom underwent volumetric MRI at baseline and 1-year, the TIA group had a higher mean annualized percentage atrophy rate at 0.82% versus 0.33% in the controls [90]. Both diastolic blood pressure and white matter disease severity were correlated with the atrophy rate in TIA patients. In a study of TIA/minor stroke patients from the Extended-CATCH (CT and MRI in the Triage of TIA and minor Cerebrovascular events to identify high-risk patients) study compared to healthy controls from the Alzheimer’s Disease Neuroimaging Initiative (ADNI), patients with TIA/minor stroke demonstrated a higher hippocampal atrophy rate than controls over a 3-year interval [91]. The annual percentage change of the left hippocampal volume was 2.5% (78 mm^3^ per year) for TIA/minor stroke patients compared to 0.9% (29 mm^3^ per year) for controls; the annual percentage change of the right hippocampal volume was 2.5% (80 mm^3^ per year) for TIA/stroke patients compared to 0.5% (17 mm^3^ per year) for controls. Patients with higher annual hippocampal atrophy were more likely to report higher TMT-B times (worse performance), lower Rey–Osterrieth Complex total score and figure recall scores, and lower California Verbal Learning Test-II total recall scores longitudinally. In another analysis of Extended-CATCH and ADNI [92], TIA/minor stroke patients demonstrated a significantly higher whole brain atrophy rate than healthy controls over a 3-year period, with diabetes independently predicting a higher atrophy rate across groups. Higher atrophy was associated with worse performance on processing speed tasks (including digit-symbol coding) but was not associated with memory or executive function composite scores or individual cognitive tests for language [88].

Other clinical and radiological factors have been identified in other studies. In a prospective cohort of 92 patients in a single-center study in Calgary, Canada, patients with TIA/minor stroke (54% TIA) underwent neuropsychological testing 90-days post-event using a modified version of the NINDS-CSN VCI battery, including tests of attention, psychomotor processing speed, executive function, language, verbal/non-verbal memory, and visuospatial construction, in addition to an assessment of depression using the CES-D [93]. Lower executive function and psychomotor processing speed scores were independently associated with previous cortical infarcts on MRI, disability on the modified Rankin Scale (post-event score > 1), and depressive symptoms (CES-D ≥ 16), with processing speed scores also being associated with bilateral acute DWI lesions.

Vascular risk factors, in general, also correlate with cognitive performance in TIA survivors. In a case-control study of 68 patients with TIA and 68 controls, cognitive impairment per the MoCA was associated with a number of vascular risk factors [94]. In a study of 613 patients (123 minor strokes, 175 TIA, and 315 mimics) using phone interview assessments at three time points in the first year post-event, performance on cognitive testing was accounted for by the presence of heart failure, myocardial infarction, angina, and hypertension, with executive functioning being associated with hypertension and angina [95]. Increased stroke risk [96] was also associated with poorer cognition.

More recently, studies have combined AD-relevant imaging with typical MRI imaging in order to better understand the risk of cognitive decline after TIA. A study in Hong Kong [97] compared clinical and imaging features between consecutive patients of TIA/stroke with (*n* = 88) and without (*n* = 925) incident dementia at 3 to 6 months post-event and performed positron emission tomography (PET) in 50 patients with Pittsburg compound B, a marker that binds fibrillar amyloid-beta plaques as found in AD [98]. Age, history of diabetes mellitus, the severity of WMHs, and medial temporal lobe atrophy were associated with incident dementia, and AD-like Pittsburg compound B retention was found in 29.7% of patients with dementia versus 7.7% without dementia. This work further supports the importance of chronic brain changes such as WMH, brain atrophy, and AD pathology with respect to long-term cognitive decline after TIAs.

Genetic factors are also likely to be important. For example, among OXVASC patients with TIA/minor stroke, apolipoprotein E (APOE)-ε4 homozygosity was associated with both pre- and post-event dementia up to five years of follow-up [99]. Associations were independent of cerebrovascular burden, suggesting that they may relate to increased neurodegenerative pathology (given the known association between APOE genotype and AD) [100] or vulnerability to injury (given the association of APOE genotype with stroke recovery and neuroplasticity) [101]. In a study of 84 patients with TIA/minor stroke and 28 controls in Qingdao, China, hypomethylation of RIN3 (Ras And Rab Interactor 3) was strongly associated with TIA/minor stroke and early cognitive impairment within 7 days post-event (tested using the Boston Naming Test, Auditory Verbal Learning Test, and Trail Making Tests A and B) [102]. The RIN3 gene acts as a stimulating factor to stabilize the transportation of guanine trinucleotide-bound Ras-Related Protein in Brain 5 (GTP-RAB5) to endosomes at the plasma membrane, a process associated with cellular endocytosis, and has a negative effect on endocytosis of the amyloid β-protein. As such, it is highly expressed in AD and hypomethylated in the whole blood of patients with early-onset AD [103,104].

## 8. Future Directions

Additional work is needed to better understand the cognitive profile of patients after TIA as well as their trajectories of cognitive function, including how the determinants of delayed-onset dementia (dementia that manifests months or years after TIA) differ from those of early-onset or potentially transient cognitive impairment soon after the event. Further unclear are the mechanisms underlying the association between vascular risk factors such as diabetes or atrial fibrillation and dementia risk and whether this risk can be modified by more intensive risk factor control or stroke prevention efforts. This work requires studies to pursue longitudinal high-quality brain imaging and cognitive testing, with close monitoring of secondary prevention strategies.

The generalisability of findings from predominantly White study populations, as in the OXVASC study, to other ethnicities and healthcare systems, is also unclear. There is also a need for greater care in how we select “controls” for robust comparison with TIA patients in these studies. Studies have traditionally relied on people from the community who volunteer to participate in research, to fill their control groups. However, we need to carefully consider who volunteers for these studies in terms of their vascular risk factors, TIA risk or past history, and risks for cognitive decline. Control populations may suffer from a self-selection of people with certain combinations of risk factors, such as people who are worried about their risk of cognitive decline. One way to mitigate this might be to construct large longitudinal cohorts of people with the aim of achieving samples comparable to the general population, such as the Canadian Longitudinal Study of Aging [105] or the 1946 National Survey of Health & Development cohort in the United Kingdom [106]. Others might seek to recruit different groups of people based on their estimated risk of cognitive decline; for example, the UK-based PREVENT cohort is recruiting participants at high, medium, and low risk of late-onset AD [107]. Ultimately, studies will need to strike a compromise between a large, representative population-based study design versus a more selective population with a greater granularity of assessment and phenotypic characterization.

The clarification of the extent to which characteristics of chronic brain injury (e.g., leucoaraiosis, brain atrophy) add to the prediction of cognitive decline—particularly in terms of their rate of change over time—remains an important challenge [108]. There is also a need for studies of post-TIA cognitive decline to more consistently incorporate biomarkers for neurodegenerative disease [109]. In particular, there is a need to integrate measures of SVD with those of neurodegeneration to facilitate a better prediction of cognitive decline. The ongoing Calgary-based PREVENT study is following patients with TIA and healthy survivors for 5 years, with repeating MRI brain scans and cognitive testing [110]. The study examines the early progressive rates of brain atrophy after TIA before the clinical detection of cognitive decline. By developing and optimizing high-level machine learning models based on clinical data, image-based (quantitative susceptibility mapping, regional brain, and white matter lesion volumes) features, and cerebrospinal fluid biomarkers, PREVENT aims to provide a timely opportunity to identify individuals at greatest risk of late-life cognitive decline early in the course of the disease, supporting future therapeutic strategies for the promotion of healthy aging. Novel imaging analysis techniques being developed in PREVENT and similar studies are likely to provide further insights into neuroimaging signatures of cognitive decline. One such technique is DeepHarP, an automatic segmentation tool for hippocampus delineation that uses an atlas-based approach to approximate the location of the hippocampus on T1-weighted MRI, followed by a convolutional neural network to segment the hippocampus from the identified region of interest. This method was developed and validated using multicenter datasets of patients with AD, MCI, TIA/stroke, and healthy controls and achieved better accuracy than four other established hippocampus segmentation methods used for comparison while also achieving a high test-retest precision [111]. Such novel techniques can help us achieve a much better granularity in evaluating the relationship of brain atrophy to cognitive decline as opposed to simple visual rating techniques. In addition to imaging markers of neurodegeneration, consideration should be given to evaluating more disease-specific markers such as amyloid or tau species. Whereas this previously required testing the cerebrospinal fluid, the emergence of blood-based tests for amyloid and tau is set to make this much easier [112].

Standard structural imaging sequences are likely only scratching the surface of the underpinnings of post-TIA cognitive decline. In addition to more sophisticated structural imaging sequences such as DTI, functional neuroimaging using PET or functional MRI and explorations of brain connectivity, also represent an important new horizon of enquiry for patients with TIA. A recent functional MRI study of 48 TIA patients and 41 age- and sex-matched healthy controls examined the connectivity of key regions in the default mode network in these patients. The researchers found significantly decreased functional connectivity in the left middle temporal gyrus/angular gyrus both with the medial prefrontal cortex and posterior cingulate cortex/precuneus, and significantly decreased functional connectivity among each pair of medial prefrontal cortex, left posterior cingulate cortex, and right precuneus in patients with TIA as compared with healthy controls [113]. Another study of cognitive differences (using the MoCA, Trail Making Task, and the National Institutes of Health Cognition Toolbox) and resting-state functional connectivity differences between 42 TIA/minor stroke patients (an average of 3.8 years post-event) and 20 controls found that controls performed better than patients on two measures of executive functioning and had reduced resting-state functional connectivity between the frontoparietal and default mode network [114]. Increased connectivity within the frontoparietal network was associated with faster performance in patients with minor stroke. These initial studies suggest that altered or aberrant connectivity between the hubs within key networks, such as the default mode network and frontoparietal network, may play an important role in cognitive decline, given the critical role of these networks in cognition and behavior [115].

Perfusion imaging also offers another important window into studying post-TIA cognitive decline, but few studies have examined this modality in detail. For example, a whole-brain CT perfusion imaging study of 50 patients with TIA in Chengdu, China (within 1 week of the event), accompanied by MoCA and MMSE testing within 4 weeks, found that patients with and without cognitive impairment showed prolongation of time-to-peak and mean-transit-time and a reduction in cerebral blood flow, but that these changes were significantly larger in patients with cognitive impairment [116]. Mean-transit-time correlated negatively with MoCA score.

There is also a need for a “big data, big imaging” approach to understanding post-TIA cognitive decline to improve accuracy and generalizability. Many important studies in this field have had <100 participants with TIA, and virtually none have had 1000 participants. Pooling together several study datasets, with the harmonization of study protocols, promises to be an important strategy to help create large, detailed datasets of patients with TIA. The Stroke and Cognition consortium (STROKOG) is a key step forward in this regard. STROKOG includes studies with ≥75 participants who suffered or were at risk of stroke/TIA and which are evaluating cognitive function. As of 2016, the consortium included 25 studies with 12,092 participants from five continents with follow-up duration ranging from 3 months to 21 years [117]. Whereas data harmonization remains a challenge, STROKOG will help reuse and combine international cohort data to explore patient-level characteristics and cognitive outcomes in an unprecedented manner.

Finally, there is a need for therapies to mitigate post-TIA cognitive decline. At present, targeting modifiable risk factors (Table 3) and preventing recurrent strokes or silent infarcts is the mainstay of preventing dementia in these patients [118]. The best evidence to date has come from the Finnish Geriatric Intervention Study to Prevent Cognitive Impairment and Disability, a randomized, double-blind trial of a 2-year multidomain intervention (diet, exercise, cognitive training, vascular risk monitoring) versus general health advice in mitigating cognitive decline (according to a comprehensive neuropsychological test battery) in at-risk elderly people [119]. The intervention succeeded in achieving a slower rate of cognitive decline; however, it is important to note that such an intervention is yet to show proven benefit in the TIA patient population. Developing targeted interventions for post-TIA cognitive decline truly represents the next frontier in this field.

## Figures and Tables

**Figure 1 jcm-11-02637-f001:**
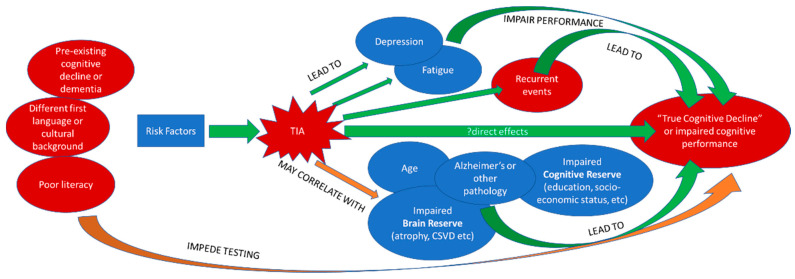
The complex interplay of potential mediators of cognitive decline after a transient ischemic attack (TIA) as well as some potential confounders.

**Table 1 jcm-11-02637-t001:** Confounding factors in the evaluation of cognitive decline after transient ischemic attack (TIA) and potential methods to address the problem in future studies.

Confounder	Potential Methods to Address the Problem
Pre-existing cognitive decline or dementia	Obtain a standardized measure of pre-TIA cognitive decline using an instrument such as the IQCODE [27]
Fatigue	Measure fatigue in participants using a standard scale such as the Chalder Fatigue Scale [68] and adjust cognitive test results for these scores (or match with corresponding controls without TIA) Consider adjusting for the time-of-day for the cognitive assessments, or standardizing the timing of the cognitive assessments
Depression	Measure depression in participants using a standard scale such as the Beck Depression Inventory-II [63], the Montgomery Asberg Depression Rating Scale [64], or the Centre for Epidemiological Studies Depression Scale (CES-D) [69,70] and adjust cognitive test results for these scores (or match with corresponding controls without TIA)
Attrition	Use multiple follow-up methods (telephone or virtual visit, home visits, and primary care records) to ascertain long-term cognitive outcomes without relying on in-person clinic visits alone
Differences in education	Adjust for years of education or highest educational milestone achieved
Patients being Untestable	Keep track of patients who are untestable in the cohort and the reason Offer cognitive testing in multiple languages where feasible Consider less culturally biased cognitive tests for multi-cultural populations (though validation in TIA or stroke may be lacking)

**Table 2 jcm-11-02637-t002:** Summary of cognitive domains reported to be affected in studies of cognitive impairment after TIA, with examples of available evidence.

Affected Domain	Example of Evidence
Overall Executive Function	- Performance on executive function tasks decreased over 6-months after TIA in a prospective cohort study [81] - TIA/stroke patients scored worse than memory clinic patients on visuo-executive subtests of the MoCA in an analysis of TIA and cognitive decline cohorts [31]
Attention and Working Memory	- 22.5-fold higher odds of impaired working memory and 6.8-fold higher odds of impaired attention versus controls in a case-control study with comprehensive neuropsychological assessment [2] - TIA/stroke patients scored worse than memory clinic patients on sustained attention subtests of the MoCA [31] - Attention improved over 6-months of follow-up (versus within 7 days post-event) in a prospective cohort [81]
Processing Speed	- 7.1-fold higher odds of impaired processing speed versus controls in a case-control study [2]
Visual and Verbal Memory	- TIA subjects had lower scores on visual and verbal memory tests compared to non-TIA subjects on domain-specific neuropsychological testing in a case-control study [85]
Verbal Fluency	- TIA/stroke patients scored worse than memory clinic patients on verbal fluency subtests of the MoCA [31]
Cognitive-Motor Integration	- Impairments in motor cognitive-motor integration in more than half of TIA patients within 2-weeks of event with more than a quarter having deterioration in performance by 1-year in a small cohort [84]

**Table 3 jcm-11-02637-t003:** Major categories of modifiable risk factors for TIA to identify and address in order to prevent recurrent events and mitigate associated cognitive decline [120].

Risk Factor Category	Examples
Major cardiovascular	Paroxysmal and/or chronic atrial fibrillation Soft or calcific carotid plaque Uncontrolled arterial hypertension
Metabolic	Diabetes Dyslipidemia Metabolic Syndrome Chronic Renal Failure
Hematologic	Antiphospholipid antibody syndrome Hyperhomocysteinemia
Infectious	Human immunodeficiency virus Syphilis
Congenital	Patent Foramen Ovale, other congenital heart diseases Dural Arteriovenous Malformations Fabry Disease
Behavioral risk factors	Smoking Cocaine and other substance abuse

## Data Availability

No original data reported.

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
