# Peer review of "The Cognitive Sequelae of Transient Ischemic Attacks—Recent Insights and Future Directions"

_jcm, 2022, doi:10.3390/jcm11092637_

Round 1

Reviewer 1 Report

  1. There are a lot of abbreviations in this manuscript. It should be better to have a list of abbreviations. Words that appear only once don't need an abbreviation.
  2. Line 280-281; which group showed the data? Patients with TCI? Please clarify.
  3. Line 299-304, 307-309; it would be more informative to mention the timing of the assessment or the testing(e.g. 3months after TIA)
  4. line 374; WMH already appeared in the previous part.
  5. Line 374-376; it would be more informative to mention the discussion of the results.
  6. Line 395-400; please cite the reference.
  7. Line 449-452' please cite the reference.
  8. What do 'RIN3' and 'GTP-RAB5' stand for?

Author Response

  1. There are a lot of abbreviations in this manuscript. It should be better to have a list of abbreviations. Words that appear only once don't need an abbreviation.

Response: We appreciate the comment and have now included a list of abbreviations at the beginning of the paper. We have also made efforts to avoid the use of abbreviations for terms appearing only once in the paper.

  1. Line 280-281; which group showed the data? Patients with TCI? Please clarify.

Response: These data apply to all three groups (TCI, persistent MCI, and no cognitive impairment). We have reworded the sentence as follows: “In all three groups, Systolic systolic BP blood pressure fell between baseline and 1 month (mean reduction 14.01 ± 21.26 mmHg) while end-diastolic velocity and mean flow velocity increased (mean increase 2.42 ± 6.41 and 1.89 ± 8.77 cm/s, respectively)”.

  1. Line 299-304, 307-309; it would be more informative to mention the timing of the assessment or the testing (e.g. 3months after TIA)

Response: We have now made the timing clear as follows:

“In a case-control study of 107 patients with TIA and no prior stroke (63% women, mean age 56.6 years, tested within 3 months of the event) versus 81 controls (56% women, mean age 52.9),2 the patients with TIA performed worse on all cognitive do-mains in a comprehensive neuropsychological assessment, except episodic memory. In particular, they had a 22.5-fold higher odds of impairment in working memory, 6.8-fold higher odds of impairment in attention, and 7.1-fold higher odds of impairment in processing speed. Over 35% of patients with TIA had impairment in one or more cognitive domains within 3 months of their event, with impairment defined as a domain z-score <-1.65 (derived based on the mean of the control group). In another study if of 39 consecutive patients with carotid occlusion and ipsilateral TIAs (tested 2-6 months after the event, median 54 days) and 46 healthy controls, all of whom underwent neu-ropsychological testing, 54% of the patients were cognitively impaired.80

  1. Line 374; WMH already appeared in the previous part.

Response: We now use the abbreviation WMH here as suggested.

  1. Line 374-376; it would be more informative to mention the discussion of the results.

Response: We have now mentioned the results as suggested:

“In OXVASC, among 566 patients with TIA or minor stroke, WMH volumes and MRI measures of white matter disruption (mean diffusivity and fractional anisotropy) were strongly associated with cognitive status in patients aged ≤80 years (all p<0.001) but not in those >80 years (not significant for any of these measures).23 In particular, lower MoCA scores were associated with frontal WMH in patients ≤80 years but not >80 years.”

  1. Line 395-400; please cite the reference.

Response: Our apologies for the oversight; we have cited the following reference here:

Arba F, Quinn T, Hankey GJ, et al. Cerebral small vessel disease, medial temporal lobe atrophy and cognitive status in patients with ischaemic stroke and transient ischaemic attack. Eur J Neurol 2017; 24: 276-282. 2016/11/20. DOI: 10.1111/ene.13191.

  1. Line 449-452; please cite the reference.

Response: These lines were referring to the results described in the previous paragraph and were meant to be “placeholders” during the manuscript drafting process. They were erroneously retained in the submitted draft and we have corrected this error.

  1. What do 'RIN3' and 'GTP-RAB5' stand for?

Response: RIN3 stands for Ras And Rab Interactor 3. GTP-RAB5 stands for Guanine trinucleotide-bound Ras-Related Protein in Brain 5. We have included these expansions in the paper when we mention these terms. 

Reviewer 2 Report

Very interesting paper. We do need more studies and , if possible, treatment to prevent post-TIA cognitive decline. If we can detect rapidly the patients that are at risk for cognitive decline, than it will be easier to treat them. 

Author Response

We thank the Reviewer for their encouraging comments.

Reviewer 3 Report

This comprehensive review by Ganesh et al. is well written and organized. No doubt, it would focus on cognitive sequelae of TIAs. To keep the readers closer to the topic, this Reviewer would suggest to add a Table focusing on the major cardiovascular (paroxysmal  and/or chronic atrial fibrillation, soft carotid plaques embolization, uncontrolled arterial hypertension), metabolic (diabetes, dyslipidemia, metabolic syndrome, chronic renal failure) congenital (i.e., dural arterio-venous malformations) and acquired (smoking, cocaine)  risk factors for TIAs occurrence. Moreover, a box with the heading "Risk Factors" should be filled in the left part of the Figure, with an arrow toward TIA, in order to focus on pathogenetic mechanisms leading to TIA, which I suggest to report in the new Table.

Author Response

This comprehensive review by Ganesh et al. is well written and organized. No doubt, it would focus on cognitive sequelae of TIAs.

Response: We thank the Reviewer for their encouraging comments.

To keep the readers closer to the topic, this Reviewer would suggest to add a Table focusing on the major cardiovascular (paroxysmal  and/or chronic atrial fibrillation, soft carotid plaques embolization, uncontrolled arterial hypertension), metabolic (diabetes, dyslipidemia, metabolic syndrome, chronic renal failure) congenital (i.e., dural arterio-venous malformations) and acquired (smoking, cocaine)  risk factors for TIAs occurrence.

Response: We have included this as Table 3 (lines 582-583):

“Finally, there is a need for therapies to mitigate post-TIA cognitive decline. At pre-sent, targeting modifiable risk factors (Table 3) and preventing recurrent strokes or silent infarcts is the mainstay of preventing dementia in these patients.”

Table 3. Major categories of modifiable risk factors for TIA to identify and address in order to prevent recurrent events and mitigate associated cognitive decline120

Risk Factor Category

Examples

Major cardiovascular

·        Paroxysmal and/or chronic atrial fibrillation

·        Soft or calcific carotid plaque

·        Uncontrolled arterial hypertension

Metabolic

·        Diabetes

·        Dyslipidemia

·        Metabolic Syndrome

·        Chronic Renal Failure

Hematologic

·        Antiphospholipid antibody syndrome

·        Hyperhomocysteinemia

Infectious

·        Human immunodeficiency virus

·        Syphilis

Congenital

·        Patent Foramen Ovale, other congenital heart diseases

·        Dural Arteriovenous Malformations

·        Fabry Disease

Behavioral risk factors

·        Smoking

·        Cocaine and other substance abuse

Moreover, a box with the heading "Risk Factors" should be filled in the left part of the Figure, with an arrow toward TIA, in order to focus on pathogenetic mechanisms leading to TIA, which I suggest to report in the new Table.

Response: We have added this modification to the Figure as suggested.
